# Precarity at the Margins of Malaria Control in the Chittagong Hill Tracts in Bangladesh: A Mixed-Methods Study

**DOI:** 10.3390/pathogens9100840

**Published:** 2020-10-14

**Authors:** Mohammad Abdul Matin, Nandini D. P. Sarkar, Ching Swe Phru, Benedikt Ley, Kamala Thriemer, Ric N. Price, Koen Peeters Grietens, Wasif Ali Khan, Mohammad Shafiul Alam, Charlotte Gryseels

**Affiliations:** 1International Centre for Diarrhoeal Disease Research, Bangladesh (icddr,b), 68, Shaheed Tajuddin Ahmed Sarani Mohakhali, Dhaka 1212, Bangladesh; abdul.matin@icddrb.org (M.A.M.); ching.swe@icddrb.org (C.S.P.); wakhan@icddrb.org (W.A.K.); shafiul@icddrb.org (M.S.A.); 2Department of Public Health, Institute of Tropical Medicine (ITM), Nationalestraat 155, 2000 Antwerp, Belgium; nsarkar@itg.be (N.D.P.S.); kpeeters@itg.be (K.P.G.); 3Global and Tropical Health Division, Menzies School of Health Research and Charles Darwin University, Rocklands Drive Casuarina, Darwin Northern Territory 0810, Australia; benedikt.ley@menzies.edu.au (B.L.); Kamala.Ley-Thriemer@menzies.edu.au (K.T.); ric.price@menzies.edu.au (R.N.P.); 4Centre for Tropical Medicine and Global Health, Nuffield Department of Clinical Medicine, University of Oxford, Oxford OX3 7LG, UK; 5Mahidol-Oxford Tropical Medicine Research Unit, Faculty of Tropical Medicine, Mahidol University, 420/6 Rajvith Road, Tungphyathai, Bangkok 10400, Thailand

**Keywords:** malaria, vivax malaria, health seeking behaviour, mixed methods, radical cure

## Abstract

Bangladesh has achieved significant progress towards malaria elimination, although health service delivery for malaria remains challenging in remote forested areas such as the Chittagong Hill Tracts (CHT). The aim of this study was to investigate perceptions of malaria and its treatment among the local population to inform contextualized strategies for rolling out radical cure for *P. vivax* in Bangladesh. The study comprised two sequential strands whereby the preliminary results of a qualitative strand informed the development of a structured survey questionnaire used in the quantitative strand. Results show that ethnic minority populations in the CHT live in precarious socio-economic conditions which increase their exposure to infectious diseases, and that febrile patients often self-treat, including home remedies and pharmaceuticals, before attending a healthcare facility. Perceived low quality of care and lack of communication between Bengali health providers and ethnic minority patients also affects access to public healthcare. Malaria is viewed as a condition that affects vulnerable people weakened by agricultural work and taking away blood is perceived to increase such vulnerability. Healthcare providers that initiate and sustain a dialogue about these issues with ethnic minority patients may foster the trust that is needed for local malaria elimination efforts.

## 1. Introduction

Despite substantial progress in reducing the burden of malaria, an estimated 3.2 billion people remain at risk globally [1]. Outside of sub Saharan Africa, 70% of the global burden of malaria is in the Asia-Pacific, with 1.8 billion people at risk for malaria [2]. In Bangladesh, across 13 endemic districts bordering India and/or Myanmar, an estimated 10,523 cases were reported in 2018 and an estimated 17,225 cases in 2019 [3,4]. Since 2007, Bangladesh’s National Malaria Control Programme and its community-based activities have been implemented in partnership with the Bangladesh Rural Advancement Committee (BRAC), an international development organization that leads a consortium of 20 non-governmental organisations (NGO) supervised by the Directorate General of Health Services [5,6]. The programme invests in both preventive and curative malaria measures. The former includes distribution of Long-Lasting Insecticide Nets (LLIN) and Insecticide Treated Nets (ITN), intermittent insecticide residual spraying, and awareness building programmes. The curative interventions include presumptive case management, early diagnosis and prompt treatment, and referral of complicated cases to tertiary facilities [5,6]. As a result, malaria prevalence decreased by 65% from 2007 to 2012 [5,7,8], and by 81% from 2010 to 2018 [9]. Nevertheless, transmission remains high in the Chittagong Hill Tracts (CHT), an area from which more than 90% of all national malaria cases are reported [9,10,11]. The highest malaria prevalence of the CHT can be found in Bandarban district, contributing about 60% of the total number of confirmed cases in Bangladesh since 2017 [9]. The ethnic minorities inhabiting this region practice a type of slash and burn cultivation, called “Jhum” in local terminology [11,12], which is a risk factor for malaria due to the increased exposure to infectious bites inherent to the working and sleeping in or close to the forest [12,13,14]. Ethnic minority populations, such as the hill tribes of the CHT, are disproportionately affected by forest malaria in Bangladesh and more generally across South-East Asia [15,16,17,18,19,20]. Indeed, despite significant improvements in LLIN/ITN coverage in Bangladesh [2,21], many challenges remain, including the extension of malaria health service delivery to such remote highland areas populated by ethnic minorities [7,21,22].

Effective malaria health service delivery is further complicated by the rising proportion of malaria due to *P. vivax* over the last decade [23,24]. This complication is attributable to the ability of the parasite to form dormant liver stages (hypnozoites) which can relapse weeks to months after an initial infection. The radical cure of *P. vivax* requires treatment with drugs that kill both the blood and liver stages [25]. Primaquine (PQ) is currently the only widely available drug that eliminates *P. vivax* hypnozoites from the human host and, while it is generally well tolerated, it can cause severe side effects (haemolysis) in individuals with the inherited enzymopathy glucose-6-phosphate dehydrogenase (G6PD) deficiency [25]. Although the PQ-based radical cure is part of the Bangladeshi national treatment guidelines, its implementation is undermined by several key factors, including the lack of widely available testing for G6PD deficiency, the lack of paediatric formulations of PQ, and concerns by healthcare providers regarding PQ induced haemolysis [13,26]. Among the ethnic minorities of the CHTs, G6PD deficiency prevalence was found to be 9.0% on average, but varied between 2% to 26% among the different ethnicities [27].

The challenges of implementing radical cure of *P. vivax* are not specific to Bangladesh and are apparent across the Asia-Pacific, South America and Horn of Africa [23]. To address these challenges effectively, an understanding is needed of the social and cultural context within which current clinical practice occurs, and in which the implementation of robust G6PD testing and PQ prescription can be achieved. This has particular relevance in the CHT with multiple ethnic minority groups, whose cultural practices and socio-economic reality diverges from the majority populations national-level healthcare regulations and clinical guidelines are often developed for [16]. This mixed methods study therefore explored perceptions of the different types of malaria and associated health seeking itineraries among the ethnic minority populations of the CHT to inform an effective implementation strategy for safe and effective radical cure for vivax malaria, including the use of G6PD deficiency testing.

## 2. Materials and Methods

### 2.1. Study Site and Population

This study collected data using the Bandarban Health and Demographic Surveillance System (HDSS) of the International Centre for Diarrheal Disease Research, Bangladesh (icddr,b) [28]. This HDSS maintains registration of births, deaths, and migrations, in addition to conducting periodical census. The area covers 179 km and includes 107 hill tribe villages (called *para*), 2 unions (administrative unit of local government) and 4296 households. If residents experience malaria-like symptoms they can contact a field worker from the HDSS, who will do a malaria rapid diagnostic test (RDT) and provide free treatment if positive while recording the malaria episode.

The majority of Bandarban residents identify with one of twelve different hill tribes, with a minority of Bengali, Bangladesh’s main ethnic group. The hill tribe populations live predominantly in forested or rural areas, and daily labour and agriculture are the most common occupations. This includes the practice of slash and burn cultivation, where crops are grown on remote, steep hillsides [11].

Healthcare is organized at different levels: at village level there are village health workers known as “shasthya shebika” (supported by BRAC), informal and formal drug outlets, traditional healers (“boidhyo”), a public health centre; at union level there is a public family welfare centre, private clinic, private hospital, government hospital, and the icddr,b malaria surveillance facility.

The study population included community members from the hill tribe villages as well as healthcare workers from all these healthcare modalities, including doctors, nurses, lab technicians, health assistants, village health workers, traditional healers, private practitioners and informal drug vendors.

### 2.2. Study Design

A mixed-methods study was carried out from May 2018 to May 2019, following a sequential exploratory design comprising two consecutive research strands. In standard annotation, this can be presented as follows: [QUAL
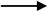
QUAN] [29], whereby the preliminary results of the qualitative strand informed the development of the structured questionnaire used in the quantitative strand. This mixed methods study aimed to identify potential community-level barriers and enablers for future roll-out of G6PD deficiency testing and radical cure. The mixed methods study was embedded in a larger case-control study assessing the relationship of G6PD activity and malaria infection. The case-control study enrolled participants under surveillance since the start of the HDSS in 2009 [28]. All participants were free from malaria at time point of enrolment and matched for sex, age, place of residence, and duration of being under surveillance. Cases had had at least one malaria episode registered in the surveillance system and controls had no recorded history of malaria.

#### 2.2.1. Qualitative Strand

Data collection: One Bengali social scientist (M.A.M.) collected data, local staff translated for informants who did not speak Bengali. Interviewing and observations were used as data collection techniques. In-depth interviews were held face-to-face, guided by a continuously adapted interview guide in line with an iterative design. Informants were classified according to relevant variables such as gender, age, history of malaria, subsistence strategies, locality, etc. to allow for internal variation and comparison. Informal interviews, not recorded by a voice recorder, with people in the villages and at the health centres constituted an important part of the data collection process. The observation of people’s behaviour is also a fundamental part of qualitative research as employed in public health research. Thus, people’s ideas and reported behaviour were contrasted with their observed actions. Both informal conversations and observations were transcribed in field notes, and each single interview, observation and conversation constituted a separate data document to be entered in the final qualitative database.

Sampling: Sampling of informants for the qualitative research was theoretical, referring to the following characteristics: (i) Purposiveness: participants were chosen on purpose and not randomly; (ii) Gradual selection: participants were theoretically selected, i.e., in accordance with emerging results/theory; (iii) Maximum variation: i.e., informants, who provide contradictory information, were systematically included in the sample. Respondents were theoretically sampled at community level, at public and private health facility level, and informal and traditional care provision level.

Analysis: Qualitative data collection and analysis were concurrent and data analysis was therefore an iterative process. Preliminary data collected through different techniques were intermittently analysed in the field after which further investigations were conducted confirming or refuting temporary results through constant validity checks until saturation was reached. Raw data was processed and coded inductively to generate and/or identify analytical categories or themes for further analysis. Abductive analysis involving the iterative testing of theoretical ideas, was used to refine and categorize themes grounded in the data. ATLAS.ti Scientific Software Development GmbH version 5.2 was used for the systemization and thematic analysis of qualitative data.

#### 2.2.2. Quantitative Strand

Data collection: A questionnaire was developed, based on the emergent qualitative findings, and was administered to participants in the ongoing case-control study. The questionnaire focused on perceptions of *P. vivax* and other malaria infections, malaria health seeking itineraries, acceptability of G6PD testing, and adherence to malaria treatment. For participants <18 years old, their legal guardians were asked to answer for them.

Sampling: An individual random sample of 100 participants was drawn from all the cases included in the larger ongoing case-control study; another 100 participants were randomly selected from the controls included in the case-control study (total sample size of 200). A total of 85 participants per group were required in order to detect a significant difference in responses between cases and controls with 95% confidence and 80% power. An additional 10% of respondents was added, assuming compromised data in at least 10% of all cases due to non-response.

Data analysis: EpiData was used for data entry. Descriptive analysis was carried out in IBM SPSS Statistics Version 25. A chi-square test was carried out to determine whether being a case (respondent with a malaria history) or a control (respondent with no malaria history) was significantly associated with any of the health seeking behaviours identified in the survey.

### 2.3. Ethical Approvals

The protocol was reviewed and approved by the Human Research Ethics Committee (HREC) of the Northern Territory (NT) Department of Health and Menzies School of Health Research (HREC 2017-3010), the ethical review committee of the icddr,b (PR-18001), and the institutional review board of the Institute of Tropical Medicine (1225/18). Written informed consent, and assent in the case of minors above the age of 11 years, was obtained from all participants and their legal guardians. For the qualitative strand, oral consent was collected, documented by the researcher in the presence of a witness.

## 3. Results

### 3.1. Participant Demographic Details

A total of 40 in-depth interviews, 11 informal interviews, 14 observations and 16 informal group discussions were conducted in the qualitative strand of the mixed methods study (Table 1).

Respondent characteristics for the quantitative survey strand are displayed in Table 2. There was no significant difference in the demographic profiles between cases and controls because these were matched for place of residence and respondents of the same villages tend to be of the same ethnicity and similar socio-economic status.

### 3.2. Setting the Stage for Malaria Elimination Activities

#### 3.2.1. Precarity

Hill tribe communities live in precarious socio-economic environments. The majority of survey respondents did not own the land they live and/or work on (61.5%). Subsistence work, including overnight stays on Jhum sites or in the forest exposes them to malaria vectors more frequently than other populations in Bangladesh, while limited access to clean water and sanitation (38.5% had no access to toilet facilities, see Table 3) increases their exposure to other infectious diseases. When they sleep in the village, their housing structures provide limited protection from insects and nuisance; the majority of respondents had housing structures with bamboo walls (67.0%), mud floors (50.5%) and a tin roof (98.5%). Nevertheless, 92.0% of respondents report to have slept under a bed net the night before the survey, although the qualitative interviews suggest these numbers may not be accurate and could potentially be explained by a social desirability bias. All respondents except one used wood as cooking fuel inside housing structures. Almost 39.0% of respondents had no access to toilets, while 32.0% had access to a pit latrine without lid, and 26.5% to a pit latrine with a lid. The main economic capital that hill tribe communities had was their animals (mainly poultry, cows, goats, and pigs) which they kept in and around the house (Table 3).

#### 3.2.2. Perceptions of Malaria Etiology and Vulnerability

“Malaria comes from the weather and the surrounding environment, because we have to drink dirty water of different streams during our work in the forest where mosquitoes lay eggs.” (Adult male farmer)

In the past, people from local communities considered malaria to be caused by a polluted wind. Nowadays many people still consider malaria to be some kind of pollution; a poison transmitted through polluted water or mosquito and other insect bites. Water is perceived to become polluted mostly in the forest where mosquitoes lay eggs and leaves fall in the water and rot. The weather is also perceived to influence infection; working in the rain, in the fog, or in very hot weather is believed to trigger malaria episodes. Almost half of survey respondents (46.6%) report weather-related fevers to be the most common fevers in the community (see Table 4).

However, it is only people with “weaker” bodies that are perceived to be vulnerable to these conditions. Strong bodies continuously fed by nutritious food are believed to be able to resist malaria. This perceived vulnerability entails an assessment of one’s socio-economic status, as the subsistence work that is required on Jhum sites makes people unable to take the rest, food and hygiene that is needed to make a body strong.

When asked about the cause of malaria through an open question in the survey, without prompting potential answers, the majority of respondents reported not to know. When prompting, meaning that enumerators actively listed all possible causes identified through the qualitative study and presented in Table 5, respondents confirmed that many of these causes could cause malaria: 90.7% believed malaria to be caused by mosquito bites, 61.8% by other jungle insects, and 51.0% by having a dirty house. Mosquito and other insect bites are considered almost unavoidable, as during the malaria peak season people often stay overnight at their Jhum site or in the forest. When they are not staying at the Jhum site or going hunting and logging in the forest, outdoor socializing is the norm at the end of the day. Activities entail visiting outdoor tea stalls, shops or places with TV’s in the village. Whether at Jhum sites or during outdoor gatherings, impregnated bed nets are impractical to use at those times.

#### 3.2.3. Perceived Malaria Types and Symptoms

“I know what malaria is, it creates immense problems in the human body, it damages the cognitive development of the human body. I had experience with this when my son passed away, affected with cerebral malaria, 7 years ago.” (Elderly male village leader)

Malaria was perceived to mostly cause fever, chills, nausea, fatigue and body pains, sometimes further causing brain problems. Informants shared their views that when the full course of malaria medication is not finished then there is a significant chance the human body becomes re-infected. “Old germs” are also perceived to be able to come back 1–2 years later, becoming triggered by mosquito bites, especially by the more “potent” bites in the forest. Informants noted that they are not able to explain clearly what and how “germs” cause malaria symptoms, yet it is clear to most that symptoms are an indication of the presence of malaria. Only 3.0% confirmed that one can have malaria without clear symptoms (Table 6). In terms of risk, living in a highland area and staying in the forest are perceived to be associated with a higher risk of malaria.

These perceptions did not translate into knowledge of different malaria types, as only 16.5% of survey respondents had heard about different types of malaria (referring to different Plasmodium species). Among those who reported to have had malaria before, the majority did not know what malaria species had caused their most recent malaria episode (Table 6).

### 3.3. Factors Driving Health Seeking Itineraries

#### 3.3.1. Illness Perceptions

“Suppose someone went to take a bath besides the waterfall; then spirits may be attracted, and the person gets affected with fever. In that case you must pay worship, otherwise your fever will not be cured by medicines. For this reason, people go to traditional healers (“boidhyo”) and then pay worship to spirits”. (Adult male farmer)

When spiritual powers are perceived to be causing someone’s illness, people rely on traditional healers called “boidhyo”. In these cases, pharmaceuticals are thought to be ineffective and consulting with medical practitioners pointless. The boidhyo diagnoses the illness based on the patient’s age, previous history and lunar calendar, and further recommends and arranges worship and/or sacrifice with different types of wild and domestic animals, bathing in streams and/or staying in the deep forest. In addition, the boidhyo further prescribes dietary restrictions (for example, no pork consumption during perceived malaria episodes) and traditional medicines (mostly various forest roots and leaves), which are less costly than pharmaceuticals and despite their perceived prolonged activity, widely considered effective for various symptoms. However, this knowledge is perceived to be eroding because it takes increasingly long to find and collect the essential roots and leaves from the forest, and thus preparing the right dosages becomes a lengthy and arduous process. In the quantitative survey, people did not report consulting with the boidhyo for malaria symptoms (see Table 7), however, for fever this health seeking avenue frequently came up in informal interviews.

#### 3.3.2. The Importance of Blood

Blood taking was considered a serious issue, as blood is perceived to be an essential and irreplaceable element in the human body that provides energy for work. If someone becomes injured, the perceived primary goal is to stop the bleeding as early as possible, making use of traditional leaves, roots or medicines. Providing blood for a diagnostic test, therefore, can be perceived to reduce one’s strength, which is needed to perform the harsh agricultural work while coping with variability in climatic conditions (both cold and hot weather). One informant mentioned that she “had provided a blood sample for the diagnosis of malaria. I don’t care about providing a blood sample from my finger, but I am afraid about providing blood from my arm or vein. It makes me afraid because more blood goes away through the vein or arm, that is the main cause of fear.” (Adult female homemaker)

Taking blood from someone is therefore a precarious activity and should only be done by skilled and careful healthcare workers. A community will not give blood without the community leader’s consent to a study requiring the blood sampling. The village leader is charged with disseminating the correct information about the need for blood taking among the community members.

“I cannot create a single drop of blood myself. Blood is the gift of God, I cannot invent it, so it is required to use carefully.” (Adult male farmer)

The amount of blood required for the venous sampling in the case-control study to identify the levels of G6PD activity was considered problematic as a larger volume than a regular malaria diagnosis was collected. This needed to be well negotiated between the research team and the village leader.

“It is not really equal to 5 fingers of a hand, so some people may not agree to provide the blood sample, it depends on one’s intention, so it is required to make them understand the importance of the study.” (Elder male village leader)

The perceived importance of blood also relates to people’s interest in knowing about the results of the laboratory tests. Receiving laboratory reports was reported to empower people to take charge of their health seeking itineraries as well as to impact on future decisions for blood sampling. One respondent narrated that “a research team had collected blood from my wife few months before, but they didn’t provide us with the lab report. Now my wife feels joint pain, is not able to perform hard work and carry even a 10-kg load, whereas she could work a long time in Jhum earlier and carry heavy loads on her back. This just happened after providing blood.” (Elder male farmer)

The importance of initiating a dialogue about the results was highlighted by the study population in the context of this study in particular, because the amount of blood required for the venous samples was perceived to strain the body more than samples taken during routine care. Nevertheless, the perceived importance of blood can be considered to transcend its use for scientific studies only and must be considered in routine care as well.

#### 3.3.3. Social Networks and Home Treatment

When symptoms first occur, people initially share their illness first within their household, then with their neighbours and their community leader. They subsequently rely on the advice of such knowledgeable relatives and friends or the community leader, advice which often includes “waiting it out for 3 days” and sharing remaining medicines, consisting mostly of paracetamol and home remedies. For fever such home remedies predominantly include wiping the body with wet cloths and pouring water over one’s head to lower body temperature. People with more financial resources reported buying pharmaceuticals from local drug shops as a first resort. Only when no relief was experienced from these home remedies and drugs, including those shared by neighbours, would the village health worker be called upon or other health seeking itineraries initiated.

#### 3.3.4. Provider Preferences

When experiencing more severe symptoms such as joint pain, body shaking, vomiting or “not being able to move eyes”, people would consult with a healthcare worker for diagnosis. Patients would only attend a private hospital when acutely or severely unwell because it took them longer to travel to these facilities. Public hospitals were not preferred in severe cases, because the quality of care was perceived to be too low and less able to deal with emergency situations adequately. This perception about the effectiveness of public health facilities is reflected in the survey results about public health care utilization: when asked where people in their community preferred to go first when they suspected having malaria, 18.0% reported a preference for the village health worker, 11.5% the local public health centre, 5.5% the government hospital (Table 7). One informant noted that “If I go to government clinic the health care providers suggest consuming the same drugs for fever, body pain, headache, cold and cough, which is not curing the disease”—indicating little faith in government staff’s capacities and knowledge.

Informal drug vendors are preferred because they are perceived to be less expensive than public health facilities, less time consuming and easy to access beyond working hours. Patients’ initial uncertainty about the aetiology of their symptoms matches well with the wide array of treatments for each and every symptom that drug vendors have: half of respondents from the survey reported going to the market drug vendor first (49.5%), increasing to 67.5% when prompted (Table 7). Although only 19.1% mentioned the icddr,b surveillance facility as a first recourse when not prompted, people remembered the iccdr,b leading to 58.3% of first choice when prompted. Likely due to their actual previous contact with icddr,b, the difference between cases and controls in reporting this facility as first recourse as a free response was significant (*p* < 0.001). The limitation of the icddr,b surveillance facilities is that they are focused only on malaria, and people prefer to consult with providers that can prescribe for other conditions as well, as long as the perceived cause of their symptoms is still uncertain. However, when it comes to respondents who claimed to have had malaria previously, the main provider was reported to be the phone call to icddr,b in 53.3% of cases, and informal providers in 40.0% of cases (Table 8). This finding is further supported by 60.0% of respondents claiming that their diagnosis was based on a blood sample (Table 8). It should be noted, however, that the number of respondents reporting to have had malaria previously, did not match the number of respondents listed as cases in the surveillance system. Six controls reported having had malaria before, and one case reported to not have had malaria before. As we report respondents’ perceptions and behaviours here, health-seeking behaviour is presented in relation to self-reported malaria episodes.

#### 3.3.5. Accessibility of Health Facilities

Local public health facilities only serve the local population for a few hours a day, reportedly from 10am to 2pm, times that overlap with working times on the Jhum sites or in the forest and thus were perceived to be relatively inaccessible. In contrast to local public health facilities, people did have faith in the quality of care of larger and more distant urban public hospitals, but distance presented an important barrier. When taking the time to travel that far, patients usually have to stay a few days along with a caregiver. Patients with limited social networks simply do not have the human resources to hinder routine work and ensure the required caregiver’s presence at the hospital.

### 3.4. Factors Driving the Use of Medicines

#### Malaria Treatment Preferences

There was a general perception among the hill tribe populations that drugs given by local public health facilities tend to be of worse quality compared to those from private facilities. The fact that 68.1% of survey respondents prefer to go first to drug vendors when they suspect malaria, aligns well with the finding that 61.0% of survey respondents mention paracetamol as preferred treatment choice for malaria unprompted, increasing further to 82.0% when prompted (see Table 9). Although hardly anyone reports using IV drips and injections as first-choice treatments for malaria when not prompted, the majority of respondents (77.5 and 72.5%, respectively) report to do so when prompted. Informants report these treatments and administration routes are also commonly available at informal and private facilities and practitioners. None of the respondents mentioned the first-line artemisinin-based combination therapies Artefen or Coartem (first line treatment for falciparum malaria), and only one respondent mentioned chloroquine (first line treatment for vivax malaria).

In relation to the reported use of treatment during a reported malaria episode, almost all respondents said to have used tablets to swallow (97.1%), although not everyone reported to have received the different kinds of tablets that are part of the first-line treatment plan (i.e., both artemisinin-based combination therapy and single-dose primaquine, also for *P. falciparum* patients) (Table 10). The reported preference for IV infusions and injections mentioned above—which in the treatment plan are reserved only for severe cases—did not feature strongly among respondents who reported to have had malaria infections before (Table 10).

### 3.5. Factors Driving Patient Adherence to Antimalarials

Due to the preferred home-treatment described above, respondents reported an important delay in treatment seeking after symptom onset. The majority (62.9%) of respondents sought treatment only after more than 2 days of symptoms (Table 11). Informants narrated that they are not usually informed by local health professionals about side effects of antimalarials. However, they also expressed little desire to know of the inconveniences in advance, as the medicines would have to be consumed anyway. Some informants did mention that they felt hot and dizzy and experienced headaches and could not perform routine work due to the medication. These side-effects would prompt the perceived need to eat nutritious food and take rest, but this would usually be unaffordable for the hill tribe populations. Many informants added that maintaining the schedule of the malaria treatment regimen is difficult, because they often forget to take treatment due to the harsh work requirements. Most informants in the qualitative study stated they would prefer to take the medicine in the evenings after work as that would not interfere with direct work requirements. Although the qualitative study found many such expressions of difficulty to adhering to different anti-malarial regimens, in the survey 97.2% respondents reported to have received anti-malarial tablets during their reported last malaria episode (N = 105), among whom 98.1% report to have finished the entire course (see Table 11). Similar to other behaviours that are expected by public health authorities, such as consistent use of mosquito nets (see above), self-reported adherence to malaria medication may also be influenced by a social desirability bias.

## 4. Discussion

This mixed methods study provides important insights into the perceptions of malaria and associated treatment seeking behaviour amongst the hill tribe populations in the CHT [12]. Implementation strategies for revised malaria diagnostic and treatment policies should be contextualized and consider the following observations.

Firstly, hill tribe populations live in precarious socio-economic conditions which increase their exposure to malaria, especially through labour-intensive subsistence work at Jhum sites and in the forest, which also complicates timely access to appropriate malaria diagnosis and treatment. Secondly, febrile patients in Bandarban often self-treat, including consuming herbal home remedies and watering therapies. They also consult with traditional healers and local leaders, before attending a healthcare facility. Public health care facilities are not well embedded within the social fabric of the hill tribe communities, whose realities are based on hard work on *Jhum* sites and in the forest, with strong social networks that provide first-hand medical advice and pharmaceuticals that are more accessible and closer to home. Previous experiences with malaria do not change such health seeking behaviour. Except for making the phone call to the malaria surveillance system when experiencing malaria-like symptoms, people who had had malaria did not report very different health seeking itineraries than those who had never had malaria before. Thirdly, people have limited knowledge of the different species of malaria parasites and associated first-line treatments. Malaria is viewed as a condition that does not trouble “strong bodies” (those who need not do agricultural work) and therefore affects only people weakened by the hard Jhum work and the lack of rest this entails, a situation which reportedly coincides with a lack of nutritious food, personal hygiene and a hygienic environment, and access to clean drinking water. Taking away blood is perceived as increasing individuals existing vulnerability to ill health, hence hill tribe communities place particular importance on blood collection as a medical intervention. Blood is considered to contain the life force which powers the body to handle the harsh Jhum work. It is therefore not straightforward for researchers and healthcare staff to negotiate a blood sample to test for untranslatable genetic conditions (e.g., G6Pd deficiency), with the aim of preventing side-effects from drugs people usually have no access to. To facilitate such negotiation, it would help if people received their test results and are engaged in a dialogue about the importance of those results, both to advance scientific knowledge and (re-)direct their own health seeking behaviour. Prior to the roll out of the additional blood taking, such as for G6PD testing, both in studies and routine care, its purpose must be well understood, accepted and promoted by prominent community leaders, and promises to deliver the test results to participants must be kept. The hill tribes in Bangladesh are not unique in their perceptions of blood. Around the world, fear of giving blood has been associated with conceptions of blood as a life-force [30,31,32], linking the lack of blood to the loss of strength and consequent weakness or illness [33,34], and even with the perception that its importance makes it a tradable commodity which needs to be financially valued [30]. Fourthly, people reported having limited access to public health care, either through its perceived low quality of care or lack of communication between Bengali health provider and hill tribe patient. Malaria in this setting is also perceived as a socio-economic condition, for which patient care is at least partly embedded in the patient’s social context. Adding on new pharmaceuticals, point of care testing, or new treatment regimens to a malaria elimination strategy that already has little bearing on ethnic minority communities’ realities will likely make little difference, as has been shown in other ethnic minority settings across South-East Asia [19,35,36,37]. The strategy itself needs to dig its roots within the social world of the malaria-affected communities it targets.

However, developing grounded and contextual malaria elimination strategies does not necessarily translate to merely implementing community engagement strategies or “empowering” community members to take charge, as it is not hill tribe people’s goal or responsibility to develop and implement public health strategies that could lead to the elimination of malaria [38,39,40]. Existing community structures (both in terms of social and traditional care) would not be able to take on additional responsibilities without the improvement of the living-, working- and other socio-economic conditions that delineate the parameters of malaria exposure for ethnic minority communities in the first place. The precarious conditions of the CHT hill tribe populations result in them having little resilience against the socio-economic consequences of infections and little time and capacity to be conducting community responses to mitigate those infections. Multi-sectoral economic, social, agricultural and health care interventions are needed to decrease such vulnerability to malaria infection and its consequences. This study also demonstrates that quantitative studies conducted without prior in-depth knowledge of the context may not be able to collect relevant data that address their aims. The differences between the unprompted, open, answers in the current survey and the prompted answers, based on locally relevant categories identified through in-depth qualitative research, was significant. Multisectoral interventions would therefore need to be based on transdisciplinary research, including triangulation of research techniques as well as the variety of community and policy stakeholders included in the research process.

Prior to such multisectoral interventions, a concrete initial action could be focused on improving the unidirectional communication between first-line healthcare workers and community members and initiate a bidirectional dialogue when moving towards a revision of the treatment guidelines [41]. This should include supporting the healthcare staff (through adequate income and education) to inform patients correctly on a number of aspects of medical management, i.e., explaining exactly what is expected of patients and why, how much blood is taken and how that affects their body, what a malaria parasite is and what treatment does to both their bodies and their capacity to work, at what times it is appropriate and inappropriate to take treatments in light of the local harsh working conditions, and to pro-actively offer insights on how to mitigate treatment side-effects and associated socio-economic hardship. Currently, however, the often-low formal education levels and cultural “otherness” of the CHT hill tribe populations drive healthcare workers to underestimate patients’ capacities and overestimate their own power to take charge of patients’ therapeutic itineraries.

### Limitations

Data collection was done by a Bengali researcher, which made language a primary barrier in communicating with informants. The fact that the Bengali researcher was considered an “outsider” further complicated trust and subsequent access to ethnic minority informants, as well as increasing social desirability bias for potentially sensitive questions regarding health seeking behaviour at public health facilities and treatment adherence. This was accommodated for by working with local HDSS staff and translators for all community-based research and engagement. The results presented about health seeking behaviour during previous malaria episodes may further be affected by recall bias, as these malaria episodes may have happened a long time ago.

## 5. Conclusions

In conclusion, this study highlights that if malaria elimination is to be achieved in remote areas with diverse ethnicities and perceptions, then the implementation of malaria control strategies need to be adapted to the local context, ensuring that culturally acceptable interventions are pursued and local communities engaged fully in the benefits these would bring.

## Figures and Tables

**Table 1 pathogens-09-00840-t001:** Participants in the qualitative strand.

In-Depth Interviews (N = 40)	N
Farmers	7
Government job	10
Private job	13
Home-based business/housewife	3
Health providers	3
Religious leader	1
Teachers	3
Informal interviews (N = 11)	
Farmers	6
Government job	2
Home-based business	2
Student	1
Observations (N = 14)	
Hospital	4
Lab	2
Health facility at community level	6
Community settings	2
Informal group discussions (N = 16)	
Among health staff	3
Among community members	13

**Table 2 pathogens-09-00840-t002:** Participant profiles in the quantitative survey.

	Controls * (N = 100)	Cases * (N = 100)	Total (N = 200)	
	n	n	n	%
Sex				
Male	52	52	104	52.0
Female	48	48	96	48.0
Religion				
Buddhism	85	85	170	85.0
Islam	12	12	24	12.0
Christianity	2	2	4	2.0
Hinduism	1	1	2	1.0
Ethnicity				
Marma	58	56	114	57.0
Tanchangya	15	17	32	16.0
Bengali	13	13	26	13.0
Khyang	6	6	12	6.0
Chakma	6	6	12	6.0
Bawm	1	1	2	1.0
Tripura	1	1	2	1.0
Occupation				
Subsistence farming	34	38	70	35.0
Student	27	30	57	28.5
Homemaker	14	11	25	12.5
Daily plantation worker	13	13	26	13.0
Construction	6	6	12	5.9
Work in rubber plantation	4	5	9	4.5
Small business owner	4	2	6	3.0
Service	2	1	3	1.5
Disabled	2	0	2	1.0
<5 years old	0	1	1	0.5
Driver	1	0	1	0.5
Seasonal plantation worker	0	1	1	0.5
Regular plantation worker	1	1	1	0.5

***** As case and control groups each comprised 100 participants, only absolute numbers are shown in these columns.

**Table 3 pathogens-09-00840-t003:** Socio-economic conditions of the hill tribe populations.

	Controls (N = 100)	Cases (N = 100)	Total (N = 200)	
	n	n	n	%
Economic capital *				
Land ownership	38	39	77	38.5
Keeps animals	80	82	162	81.0
Keeps cows	43	31	74	37.0
Keeps dogs	6	10	16	8.0
Keeps poultry	57	59	116	58.0
Keeps goats	22	27	49	24.5
Keeps cats	7	8	15	7.5
Keeps pigs	19	16	35	17.5
Keeps sheep	0	1	1	0.5
Bed net use				
Slept under LLIN last night	91	93	184	92.0
Did not sleep under net	7	7	14	7.0
Slept under untreated net	2	0	2	1.0
Access to sanitary facilities				
Access to improved toilet	0	1	1	
Access to pit latrine with lid	27	26	53	26.5
Access to pit latrine without lid	31	33	64	32.0
Access to hanging open latrine	3	2	5	2.5
No access to toilet facility	39	38	77	38.5
Use of cooking fuel				
Wood	100	99	199	99.5
Gas	0.0	1	1	0.5
Observed housing structures *				
Ground level house	70	69	139	69.5
Stilted house	30	31	61	30.5
**Walls**				
Bamboo	70	64	134	67.0
Mud	24	27	51	25.5
Wood	5	5	10	5.0
Cement	4	5	9	4.5
Tin	1	1	2	1.0
Brick	0	2	2	1.0
**Floor**				
Mud	50	51	101	50.5
Bamboo	22	23	45	22.5
Wooden	16	15	31	15.5
Cement	11	11	22	11.0
Palm	1	1	2	1.0
Tin	1	0	1	0.5
**Roof**				
Tin	98	99	197	98.5
Cement	1	0	1	0.5
Missing	1	1	2	1.0
**Windows**				
Closable windows	59	57	116	58.0
No windows	33	34	67	33.5
Open walls	7	9	16	8.0
Missing	1	0	1	0.5

* Multiple responses were possible.

**Table 4 pathogens-09-00840-t004:** Reported common types of fever in the community *.

	Control (N = 100)		Case (N = 100)		Total (N = 200)			
	Free Response	Prompted Response	Free Response	Prompted Response	Free Response		Prompted Response	
	n	n	n	n	n	%	n	%
Virus	38	78	36	79	74	37.0	157	78.5
Malaria	10	59	21	68	31	15.5	127	63.5
Weather-related fever	2	43	4	51	6	3.0	94	47.0
Influenza	10	45	9	47	19	9.5	92	46.0
Typhoid	11	35	9	33	20	10	68	34.0
Environmental pollution fever	1	11	0	7	1	0.5	18	9.0
Food-related fever	0	2	0	2	0	0.0	4	2.0
Chikungunya	2	2	2	1	4	2.0	3	1.5
Don’t know	48	1	47	1	95	47.5	2	1.0
NA	1	0	0	0	1	0.5	0	0.0

* Multiple responses were possible.

**Table 5 pathogens-09-00840-t005:** Perceived aetiologies of malaria *.

	Controls (N = 100)		Cases (N = 100)		Total (N = 200)			
	Free Response	Prompted Response	Free Response	Prompted Response	Free Response		Prompted Response	
	n	n	n	n	n	%	n	%
Mosquito bite	45	86	51	95	96	48.0	181	90.5
Jungle insects	7	62	12	60	19	9.5	122	61.0
Dirty house	10	49	9	54	19	9.5	103	51.5
Virus	6	12	2	16	8	4.0	28	14.0
Being in the rain	1	14	1	8	2	1.0	22	11.0
Bad personal hygiene	0	9	0	6	0	0.0	15	7.5
Drinking unsafe water	1	6	0	5	1	0.5	11	5.5
Fog	0	3	0	5	0	0.0	8	4.0
Parasite	0	5	1	3	1	0.5	8	4.0
Hot weather	0	0	0	4	0	0.0	4	2.0
Don’t know	43	4	37	3	80	40.0	7	3.5
Environmental pollution	0	1	0	5	0	0.0	6	3.0
Not using a bednet	5	2	3	4	8	4.0	6	3.0
Eating contaminated foods	0	4	1	1	1	0.5	5	2.5
Poison	0	2	0	1	0	0.0	3	1.5
Eating pork	0	1	0	1	0	0.0	2	1.0
Poor nutrition	0	1	0	1	0	0.0	2	1.0
Spirits	0	1	0	0	0	0.0	1	0.5
Sorcery	0	0	0	1	0	0.0	1	0.5

* Multiple responses possible.

**Table 6 pathogens-09-00840-t006:** Knowledge of malaria species.

	Controls (N = 100)	Cases (N = 100)	Total (N = 200)	
	n	n	n	%
Has heard about different kinds of malaria				
Don’t know	60	58	118	59.0
No	24	25	49	24.5
Yes	16	17	33	16.5
- Doesn’t know which types of malaria	9/16	10/17	27/33	81.8
- Has heard about severe malaria	3/16	3/17	6/33	18.2
- Has heard about falciparum malaria	0/16	1/17	1/33	3.0
- Has heard about vivax malaria	0/16	1/17	1/33	3.0
Thinks someone can have malaria without symptoms				
Don’t know	72	64	136	68.0
It could be	16	21	37	18.5
No	10	11	21	10.5
Yes	2	4	6	3.0
Thinks the same malaria infection can come back after it got better				
Don’t know	57	48	105	52.5
It could be	27	31	58	29.0
Yes	12	17	29	14.5
No	4	4	8	4.0
Reported malaria type during last malaria episode				
Reported to have had malaria before	6	99	105	52.5
- Don’t know	3/6	87/99	90/105	85.8
- Don’t remember	3/6	5/99	8/105	7.6
- Severe malaria	0/6	3/99	3/105	2.9
- Normal malaria	1/6	2/99	3/105	2.9
- Falciparum malaria	0/6	1/99	1/105	1.0

**Table 7 pathogens-09-00840-t007:** Preferences for primary providers in case of malaria suspicions (N = 200) *.

	Controls (N = 100)		Cases (N = 100)		Total (N = 200)			
	Free Response	Prompted Response	Free Response	Prompted Response	Free Response		Prompted Response	
	n	n	n	n	n	%	n	%
Market drug vendor	49	67	50	68	99	49.5	135	67.5
Phone call to icddr,b **	19	44	19	74	38	19.0	118	59.0
Government hospital	8	45	3	46	11	5.5	91	45.5
Drug shop	22	30	31	40	53	26.5	70	35.0
Village health worker	21	30	15	31	36	18.0	61	30.5
Community health centre	12	23	11	30	23	11.5	53	26.5
Private clinic	1	12	1	6	2	1.0	18	9.0
Use old medicine at home	0	5	0	8	1	0.5	13	6.5
Missionary hospital	1	6	1	3	2	1.0	9	4.5
Village leader	1	4	2	3	3	1.5	7	3.5
Experienced relative	1	3	1	1	1	0.5	4	2.0
Quantam medicare	3	3	1	2	4	2.0	5	2.5
Homeopathic clinic	0	1	0	2	0	0	3	1.5
Boidyo	0	1	0	1	0	0	2	1.0
Private practitioner	1	1	1	0	2	1.0	1	0.5
Don’t know	3	0	2	0	5	2.5	0	0

* multiple options were possible. ** significant difference between case and control *p* < 0.001.

**Table 8 pathogens-09-00840-t008:** Provider during last malaria episode (N = 105).

	n	%
Treatment provider during last malaria episode (N = 105) *		
Phone call to icddr,b	56	53.3
Informal provider (drug shop or market vendor)	42	40.0
Village health worker	4	3.8
Don’t know/remember	3	2.9
Para health post	4	2.0
Private clinic	2	1.0
Main provider’s malaria diagnostic tool during last malaria episode (N = 105)		
Took blood sample	63	60.0
Did not diagnose malaria	14	13.3
Based on symptoms	12	11.4
Rapid diagnostic test	11	10.5
Don’t remember	4	3.8
Don’t know	1	1.0

* Multiple responses possible.

**Table 9 pathogens-09-00840-t009:** Preferred primary treatments for malaria among community members *.

	Controls (N = 100)		Cases (N = 100)		Total (N = 200)			
	Free Response	Prompted Response	Free Response	Prompted Response	Free Response		Prompted Response	
	n	n	n	n	n	%	n	%
Paracetamol	61	80	61	84	122	61.0	164	82.0
IV infusion	7	74	7	81	14	7.0	155	77.5
Injections	9	72	8	73	17	8.5	145	72.5
Blister pack tablets	23	35	21	36	44	22.0	71	35.5
Watering	2	18	0	23	2	1.0	41	20.5
Massage	2	14	7	20	9	4.5	34	17.0
Syrup	3	13	2	16	5	2.5	29	14.5
Mixed medicines in a bag	0	8	2	9	2	1.0	17	8.5
Single unidentified tablets	0	5	0	8	0	0.0	13	6.5
Antibiotics	1	6	1	6	2	1.0	12	6.0
Artefen/coartem	0	3	2	9	2	1.0	12	6.0
Drinking clean water	0	7	1	5	0	0.0	12	6.0
Changing diet	0	2	0	4	0	0.0	6	3.0
Herbal treatment	0	1	1	2	1	0.5	3	1.5
Traditional medicines	0	3	0	0	0	0.0	3	1.5
Homeopathic medicines	0	1	1	0	1	0.5	1	0.5
Don’t know	20	2	17	1	37	18.5	3	1.5

* Multiple responses possible.

**Table 10 pathogens-09-00840-t010:** Reported treatment during last malaria episode (N = 105).

	n	%
Tablets	102	97.1
Different types of tablets	69	65.7
Injections	10	9.5
IV infusion	9	8.6
Spiritual worship	9	8.6

**Table 11 pathogens-09-00840-t011:** Adherence to treatment during last malaria episode.

	n	%
Self-reported malaria history (N = 200)		
No	94	47.0
Yes	105	52.5
Don’t remember	1	0.5
Delay in treatment seeking after symptom onset (N = 105)		
Sought no treatment	1	1.0
Sought treatment same day	8	7.6
Sought treatment next day	9	8.6
Sought treatment after 2 days or more	66	62.9
Don’t remember	22	20.0
Reported adherence to treatment during last malaria episode (N = 105)		
Took all the pills	103	98.1
Did not take all the pills	1	1.0
Don’t remember	1	1.0

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
