# Peer review of "Precarity at the Margins of Malaria Control in the Chittagong Hill Tracts in Bangladesh: A Mixed-Methods Study"

_pathogens, 2020, doi:10.3390/pathogens9100840_

Round 1
Reviewer 1 Report
Well written report of a study on malaria control knowledge, attitudes, behaviour of remote minority groups and possible constraints
I have no concerns with regards to methods or conclusions except that it is not clear (to me) how the "cases" and"controls" were somehow randomly (?) selected for the study
Author Response
I have no concerns with regards to methods or conclusions except that it is not clear (to me) how the "cases" and"controls" were somehow randomly (?) selected for the study
Reply: thank you for your positive review. We understand that the explanation of the random selection of participants was not clear in relation to the fact that they were also cases or controls. We have edited the Sampling in the Quantitative strand in the Methodology section to make this clearer. P4 L154-156 now reads as follows: “An individual random sample of 100 participants was drawn from all the cases included in the larger ongoing case-control study; another 100 participants were randomly selected from the controls included in the case-control study (total sample size of 200).”

Reviewer 2 Report
The authors provide in depth results from a qualitative followed by quantitative investigation designed to determine how a specific, remote population views malaria and malaria treatment (and health care in general). The goal was to gather information that could be used to better treat P. vivax in ethnic minority populations in Bangladesh (and potentially elsewhere, it would be interesting to know if other studies had been performed elsewhere that showed similar or dissimilar results). The authors determine that better communication regarding healthcare and treatment, along with access to appropriate (appropriate for the population) diagnostic/treatment options, and full community engagement are required for successful malaria control.
Minor comments:
More details about the statistics and statistical tests used would be useful. The program is mentioned and certain areas where statistical analysis was performed are called out, but no in-depth details about how exactly the results were determined. For example, in line 91-92: There was no significant difference in the demographic profiles between cases and controls. And in tables it is noted if there was significant difference between case and controls (and what does that significant difference between case and controls really tell us? More discussion on that point would be useful).
Before introducing table 2 and results, it would be useful to set up the study design briefly. This is done in the methods (line 405 – 408: “A mixed-methods study was carried out … informed the development of the structured questionnaire used in the quantitative strand”) and the abstract (lines 23-35: “The study comprised two sequential strands …the development of a structured survey questionnaire used in the quantitative strand”). But having something similar (briefly) in introduction could help provide useful context for a reader if they haven’t fully read the methods section or abstract yet.
For determining if cases had had at least one malaria episode and controls having no history of malaria, how was this determined clinically? (or perhaps that is outside scope of this study since the larger research study is mentioned, but it could be useful to provide brief explanation if possible).
The difference between free and prompted response becomes clear as one reads the narrative (and is perhaps intuitively understood by most readers), but it might be useful to briefly explain more what is meant by a free or prompted response the first time it is introduced in a table (maybe a small example?).
The majority of respondents reported finishing their entire course of drugs. This seems like an important point as non-compliance can be critical with management of certain diseases in certain areas. Is compliance with drug regimen generally good (once treatment is actually sought, provided, etc) and as such not as major of a problem as it might be in other areas?
In addition, is it likely that this answer was also impacted by a social desirability bias, similar to the bug net question? Similarly, the mention of social desirability bias was interesting and a good point. Was it thought to be prevalent and perhaps should be included as a limitation, or was it thought to not impact most answers?
Is there any importance to the keeping of livestock (or the type of livestock kept) when it comes to evaluating how best to control malaria in this population? E.g. any evidence or discussion about zooprophylaxis vs zoopotentiation? This may be outside scope, but since livestock types are mentioned, wondering if this also has any impact on how control strategies could/should be implemented or if it is generally less important than the other topics.
Were there any other important differences noted in responses from participants of different demographic groups (e.g. sex, religion, occupation) that could further help inform strategies? Are the statements made generally applicable to the population surveyed, or would certain strategies or communication about specific topics be best targeted at certain people (e.g. perhaps sex A cares more about importance of blood and sex B cares more about what treatment does to both their bodies and their capacity to work?)
Author Response
The authors provide in depth results from a qualitative followed by quantitative investigation designed to determine how a specific, remote population views malaria and malaria treatment (and health care in general). The goal was to gather information that could be used to better treat P. vivax in ethnic minority populations in Bangladesh (and potentially elsewhere, it would be interesting to know if other studies had been performed elsewhere that showed similar or dissimilar results). The authors determine that better communication regarding healthcare and treatment, along with access to appropriate (appropriate for the population) diagnostic/treatment options, and full community engagement are required for successful malaria control.
Reply: Thank you for this positive evaluation of our manuscript. To address your comment that it would be interesting to know if other studies had been performed with similar results, we have added references to the Discussion that show how other malaria elimination interventions experienced challenges because these were designed for the majority populations of the respective countries and failed to align with the interests and realities of (often impoverished) ethnic minority populations (P15-16, L461-466).
Minor comments:
- More details about the statistics and statistical tests used would be useful. The program is mentioned and certain areas where statistical analysis was performed are called out, but no in-depth details about how exactly the results were determined. For example, in line 91-92: There was no significant difference in the demographic profiles between cases and controls. And in tables it is noted if there was significant difference between case and controls (and what does that significant difference between case and controls really tell us? More discussion on that point would be useful).
Reply: Thank you for your constructive remark. We agree that not enough discussion was included to explain the lack of difference between cases and controls. We have tried to address this with several edits and additions:
(i) We have added in the Methodology section, under Quantitative Strand, more information about the statistical test used (“A chi-square test was carried out to determine whether being a case (respondent with a malaria history) or a control (respondent with no malaria history) was significantly associated with any of the health seeking behaviours identified in the survey” P4 – L154-156).
(iv) Finally, we believe that it would increase readers’ understanding of the results if they can first read about the Materials and Methods (including about the surveillance system, how cases and controls are defined, who has been interviewed about what, etc.) prior to reading the results. In many other medical and public health journals, it is common to have the Methods section before the Results. We would like to argue, also to the Editors, to allow such a structure for this manuscript as well, to improve the readability of the results.
- Before introducing table 2 and results, it would be useful to set up the study design briefly. This is done in the methods (line 405 – 408: “A mixed-methods study was carried out … informed the development of the structured questionnaire used in the quantitative strand”) and the abstract (lines 23-35: “The study comprised two sequential strands …the development of a structured survey questionnaire used in the quantitative strand”). But having something similar (briefly) in introduction could help provide useful context for a reader if they haven’t fully read the methods section or abstract yet.
Reply: indeed, we also feel that reading the Methodology section first is essential to being able to interpret the results as a reader. As stated above, we would like to argue for another structure, with Materials and Methods before the Results.
- For determining if cases had had at least one malaria episode and controls having no history of malaria, how was this determined clinically? (or perhaps that is outside scope of this study since the larger research study is mentioned, but it could be useful to provide brief explanation if possible).
Reply: All participants are part of an ongoing demographic surveillance system, so if participants experience malaria-like symptoms they can contact a field worker, who will do a malaria RDT (Falcivax, Zephy Diagnostics, India) and provided with free treatment if positive. The malaria episode is hence recorded. This is outlined in another manuscript currently under review with PLoS Med, and we have added these details in this manuscript as well in the Methods on P2-3 L92-94.
- The difference between free and prompted response becomes clear as one reads the narrative (and is perhaps intuitively understood by most readers), but it might be useful to briefly explain more what is meant by a free or prompted response the first time it is introduced in a table (maybe a small example?).
Reply: thank you for noting this. This was indeed not well explained. The first time we talk about results from open vs. prompted survey questions in the manuscript, we have explained in more detail what this means. P8 L226-229 now reads as follows: “When asked about the cause of malaria through an open question in the survey, without prompting potential answers, the majority of respondents reported not to know. When prompting, meaning that enumerators actively listed all possible causes identified through the qualitative study and presented in Table 4, respondents confirmed that many of these causes could cause malaria:…”.
- The majority of respondents reported finishing their entire course of drugs. This seems like an important point as non-compliance can be critical with management of certain diseases in certain areas. Is compliance with drug regimen generally good (once treatment is actually sought, provided, etc) and as such not as major of a problem as it might be in other areas? In addition, is it likely that this answer was also impacted by a social desirability bias, similar to the bug net question? Similarly, the mention of social desirability bias was interesting and a good point. Was it thought to be prevalent and perhaps should be included as a limitation, or was it thought to not impact most answers?
Reply: Regarding social desirability bias, this is indeed a major concern with all questions about behaviour that public health authorities expect from their target population (i.e. net use, adherence to treatment, visiting a public health provider, not believing in spiritual etiologies and associated traditional healing, etc.). We have added this to the limitations (P16 L501-503) and in the Results in the part on reported treatment adherence (P14 L41-418). When it comes to treatment compliance, this is indeed a major concern across Bangladesh for various types of medication. It has been shown that in Bangladesh patients receive prescriptions but do not properly follow it; that people often take medicines at the wrong time and in incorrect doses; and that people delay health seeking due to poor satisfaction with health care facilities generally, lack of trust in public hospitals specifically, financial issues, and geographical proximity to unregulated, untrained vendors. Some publications that show this:
- Matin MA, Khan WA, Karim MM, et al. What influences antibiotic sales in rural Bangladesh? A drug dispensers’ perspective. J Pharm Policy Pract 2020; 13: 1–12.
- Hussanin S, Boonshuyar C, Ekram A. Non-Adherence To Antihypertensive Treatment in Essential Hypertensive Patients in Rajshahi, Bangladesh. Anwer Khan Mod Med Coll J 1970; 2: 9–14.
- Ahmed S, Nasrin D, Ferdous F, et al. Acceptability and Compliance to a 10-Day Regimen of Zinc Treatment in Diarrhea in Rural Bangladesh. Food Nutr Sci 2013; 04: 357–364.
- Is there any importance to the keeping of livestock (or the type of livestock kept) when it comes to evaluating how best to control malaria in this population? E.g. any evidence or discussion about zooprophylaxis vs zoopotentiation? This may be outside scope, but since livestock types are mentioned, wondering if this also has any impact on how control strategies could/should be implemented or if it is generally less important than the other topics.
Reply: Thank you for the observation. Traditionally, people of the study area keep livestock in their house. Both case and control groups have shown similar frequencies for livestock raising (80 and 82). There is no record of livestock as zooprophylaxis for malaria or any other vector-borne diseases in Bangladesh. In this study, the purpose of presenting proportions of livestock raising is to show the economic importance of livestock rather than as a potential reservoir for human malaria. We believe that raising this issue in the manuscript is beyond its scope.
- Were there any other important differences noted in responses from participants of different demographic groups (e.g. sex, religion, occupation) that could further help inform strategies? Are the statements made generally applicable to the population surveyed, or would certain strategies or communication about specific topics be best targeted at certain people (e.g. perhaps sex A cares more about importance of blood and sex B cares more about what treatment does to both their bodies and their capacity to work?)
Reply: thank you for your comment. The study population’s main characteristic is that they identify with hill tribe ethnicities and as such differentiate themselves socio-culturally from the majority Bengali population of Bangladesh. Rather than stressing the differences within the hill tribe population, our results show that the precarious working and living circumstances often associated with belonging to an ethnic minority is the main issue that needs to be taken into account for elimination strategies. We believe this has been sufficiently addressed in the Discussion. Within the study population, the quantitative strand of the study shows no important differences in malaria perceptions or health seeking behaviours when it comes to sex. However, the qualitative strand does show that occupations that involve work in the forest (including agriculture and jhum, both performed by women and men) do increase the risk perception of contracting malaria. Working far away from village residences further complicates timely access to appropriate malaria diagnosis and treatment. We have added an additional sentence on this in the discussion, P15 L427.